# The Distribution of *Neospora caninum* Secretory Proteins in Mouse and Calf Brains

**DOI:** 10.3390/microorganisms13091970

**Published:** 2025-08-22

**Authors:** Nanako Ushio-Watanabe, Rio Fujihara, Kenichi Watanabe, Manabu Yamada, Yoshiyasu Kobayashi, Yoshifumi Nishikawa

**Affiliations:** 1National Research Center for Protozoan Diseases, Obihiro University of Agriculture and Veterinary Medicine, Hokkaido 080-8555, Japan; nanakoushio75@gmail.com; 2Kochi Seibu Livestock Hygiene Service Center, Kochi 786-0043, Japan; rio_fujihara@ken2.pref.kochi.lg.jp; 3Laboratory of Veterinary Pathology, Department of Veterinary Medicine, Obihiro University of Agriculture and Veterinary Medicine, Hokkaido 080-8555, Japan; knabe@obihiro.ac.jp (K.W.); yamadam@obihiro.ac.jp (M.Y.); kyoshi@obihiro.ac.jp (Y.K.)

**Keywords:** *Neospora caninum*, encephalitis, secretory proteins, dense granule proteins

## Abstract

*Neospora caninum*, as well as *Toxoplasma gondii*, secrete proteins that facilitate the invasion of host cells and the regulation of host immune response and metabolism. However, the localization of the secretory proteins in infected animal brains has not been studied in detail. Here, we investigate the brain and intracellular distribution of the secretory proteins in experimentally infected mice and naturally infected calves through histopathology and immunohistochemistry (IHC) to detect surface antigen 1 (NcSAG1), cyclophilin (NcCYP), profilin (NcPF), dense granule protein 6 (NcGRA6), and NcGRA7. These methods revealed that numerous tachyzoites positive for NcSAG1, NcCYP, NcPF, NcGRA6, and NcGRA7 were localized in and around the animals’ necrotic lesions, and NcGRA7 was diffusely observed in the necrotic lesions of the infected mice. Moreover, IHC revealed that NcGRA6 and NcGRA7 were distributed in the cytoplasm of infected neurons around the parasites in the infected mice and calves. This suggests that NcGRA6 and NcGRA7 might be directly related to the alteration of neuronal metabolism and activity, and that NcGRA7 might be related to the formation of necrotic lesions.

## 1. Introduction

*Neospora caninum* is a type of parasite belonging to the phylum Apicomplexa, which causes neosporosis, a systemic disease. It is similar to *Toxoplasma gondii* in terms of morphology and biology but has key differences, such as its antigenicity, virulence factors, and pathogenesis [1,2,3]. *N. caninum* primarily infects dogs, causing severe neuromuscular disease, and cattle, causing abortion and neonatal mortality [3,4,5] which lead to substantial economic losses in the beef and dairy industries [5,6].

During infection, *T. gondii* uses secretory organelles called micronemes, rhoptries, and dense granules to secrete proteins that facilitate invasion and alter host cell physiology [7,8]. Proteins secreted by dense granules, called GRAs, are distributed in the parasitophorous vacuole (PV), parasitophorous vacuole membrane (PVM), and intravacuolar network (IVN) and play important roles in the growth and development of the parasite in the host cells [9,10]. Among these GRAs, TgGRA5 and TgGRA7 are localized in the PVM and may play roles in the development and maturation of the cyst wall and membrane [11], while some GRAs, such as TgGRA6, TgGRA15, TgGRA16, TgGRA18, and TgGRA24, are transported to the host cell cytosol or nucleus [12,13,14,15,16,17,18,19,20,21]. However, the distribution and the functions of GRAs in *N. caninum* infection are not well understood.

To date, NcGRA1 [22,23], NcGRA2 [24,25], NcGRA3 [26], NcGRA6 [27,28], NcGRA7 [23,27,28], NcGRA9 [29], NcGRA14 [30], and NcGRA17 [31] have been identified in *N. caninum.* By screening 18 potential GRAs alongside cyclophilin (NcCYP) and profilin (NcPF), which are related to migration to host cell and host immune system regulation [32,33,34], NcGRA6, NcGRA7, and NcGRA14 were shown to be related to the activation of NFκB, calcium/calcineurin (NFAT), and cAMP/PKA (CRE) signaling [34], with NcGRA7 also regulating the aggregation of immunity-related guanosine triphosphatase (IRG) [35]. It has been revealed that NcCYP is distributed in the tachyzoite cytosol [33], NcPF is expressed in the apical end of the tachyzoite [32], and NcGRA6 and NcGRA7 are localized in PV and PVM [27]. Moreover, immunohistochemistry detected NcGRA7 antigens in and around the inflammatory and necrotic lesions in the brain of the infected mice [34]. However, the distribution of other secretory proteins, NcCYP, NcPF, and NcGRA6, in the brain of infected animals has not been researched. Notably, even for NcGRA7, no study has examined the relationship between the distributions of these secretory proteins and lesions in cattle. *N. caninum* surface antigen 1 (NcSAG1), non-secretory proteins abundantly expressed on the plasma membrane of tachyzoites, has been widely studied as a diagnostic marker [36,37,38,39], making it a useful reference for comparison with secretory proteins. In the present study, we focused on the distribution of NcCYP, NcPF, NcGRA6, and NcGRA7, along with NcSAG1, in the brains of experimentally infected mice and naturally infected calves, to highlight a significant gap in our understanding of the pathogenesis of *N. caninum* infection in its natural hosts.

## 2. Materials and Methods

### 2.1. Ethics Statement

Our study was performed in strict accordance with the recommendations of the Guide for the Care and Use of Laboratory Animals of the Ministry of Education, Culture, Sports, Science and Technology, Japan. The protocol was approved by the Committee on the Ethics of Animal Experiments at Obihiro University of Agriculture and Veterinary Medicine, Hokkaido, Japan (Approval number: 21-39, 21-6, Approval date: 5 July 2021, 1 April 2021). All surgeries were performed under isoflurane anesthesia with every effort made to minimize animal suffering.

### 2.2. Animals

BALB/c mice were purchased from Clea Japan (Tokyo, Japan). All mice were housed in cages (<6 mice/cage, 225 × 340 × 155 mm) containing wood chip bedding under specific-pathogen-free conditions in the animal facility of the National Research Center for Protozoan Diseases at Obihiro University of Agriculture and Veterinary Medicine. Eight-week-old female mice were used for experimental infection.

A 30-day-old and a 41-day-old calf, belonging to different daily farms, emaciated with astasia and depression, were euthanized because of poor prognosis. They were routinely necropsied and examined for pathology at the Kochi Prefectural Livestock Hygiene Center and Laboratory of Veterinary Pathology, Department of Veterinary Medicine, Obihiro University of Agriculture and Veterinary Medicine. The two calves were diagnosed as having neosporosis by immunohistochemistry.

### 2.3. Parasites

*Neospora caninum* tachyzoites (Nc1 strain) were serially passaged in Vero cell monolayers. The parasites and cells were maintained in Minimum Essential Medium (Sigma-Aldrich, St. Louis, MO, USA) supplemented with 8% (*v*/*v*) fetal bovine serum (FBS; Biowest, Nuaillé, France) and 1% penicillin–streptomycin (PS; FUJIFILM Wako Pure Chemical Corporation, Osaka, Japan) at 37 °C in humidified air with 5% CO_2_. To purify the parasites, cell monolayers were scraped, centrifuged, and then resuspended in the culture medium for each assay. Aggregated tachyzoites and host cell debris were removed by repeatedly passing the suspension through a 27-gauge needle and filtering it through a 5.0 µm pore-size filter (Millipore, Burlington, MA, USA).

### 2.4. Production of Polyclonal Antibodies Against N. caninum Proteins

Purified rabbit polyclonal antibodies against *N. caninum* proteins NcSAG1 [40], NcCYP [33], NcPF [41], NcGRA6 [40], and NcGRA7 [34] had been previously prepared. Briefly, the genes NcSAG1 (nt 89-896), NcCYP (nt 54-537), NcPF (nt 1-492), NcGRA6 (nt 130-462), and NcGRA7 (nt 1-711) were amplified with PCR from the cDNA of *N. caninum* tachyzoites and cloned into the pGEX4T-1 plasmid. The recombinant proteins were expressed as glutathione S-transferase (GST) fusion proteins in Escherichia coli strain DH5a (Takara Bio Inc., Shiga, Japan). Each recombinant protein (300 μg) in Freund’s complete adjuvant (Sigma) was injected intradermally into a female Japanese white rabbit (Kitayama Labes, Nagano, Japan) on days 0, 14, 28, and 42. IgG was purified from 2 mL of serum collected 7 days after the last immunization using a protein A chromatography column (Econo-Pac^®^ Protein A Kit, Bio-Rad Laboratories, CA, USA) according to the manufacturer’s instructions.

### 2.5. Immunofluorescence

Human Foreskin Fibroblast (HFF) cells were cultured in Dulbecco’s Modified Eagle Medium (DMEM) with 10% (*v*/*v*) FBS and 1% PS, and maintained at 37 °C with 5% CO_2_. The cells were seeded onto 8-well chamber slides (WATSON Co., Ltd., Tokyo, Japan) at a density of 2 × 10^4^ cells per well. *N. caninum* tachyzoites were added at a multiplicity of infection (MOI) of 2, and infection was allowed to proceed for 48 h before fixation. The next day, the culture medium was carefully replaced with fresh DMEM containing 10% FBS and 1% PS to remove extracellular parasites and maintain cell viability. After fixation with 4% paraformaldehyde (PFA, FUJIFILM Wako, Tokyo, Japan) for 15 min at room temperature, the samples were washed three times with phosphate-buffered saline with Tween20 (PBST) and permeabilized with 0.1% Triton X-100. After blocking with 3% bovine serum albumin (BSA, FUJIFILM), the samples were incubated with rabbit polyclonal antibodies against NcSAG1, NcCYP, NcPF, NcGRA6, and NcGRA7 (all diluted 1:500) for 1 h at 37 °C. After being washed three times with PBST, the samples were incubated with mouse monoclonal antibody against β tubulin (clone: 1D4A4; Proteintech Group, Inc., Rosemont, IL, USA) for 1 h at 37 °C. After being washed three times with PBST again, the samples were incubated with secondary antibodies against rabbit IgG (H+L) cross-adsorbed secondary antibody Alexa Fluor™ 488 and against mouse IgG(H+L) cross-absorbed secondary antibody Alexa Fluor™ 594 (Thermo Fisher Scientific, Waltham, MA, USA) for 1 h at 37 °C. After being washed three times with PBST for a third time, the chamber was carefully removed from the slide and the slide was mounted with Hard Set mounting medium with DAPI (Vector Laboratories, Mowry Ave Newark, CA, USA). Observation and image acquisition were performed using a Leica Thunder Imaging System equipped with a Large Volume Computational Clearing (LVCC) module (Leica Microsystems GmbH, Wetzlar, Germany). This system is based on widefield epifluorescence microscopy and employs computational algorithms to eliminate out-of-focus light, thereby enhancing image clarity and contrast. The LVCC method allows for the visualization of intracellular protein localization with improved resolution and minimal photobleaching, and was used as an alternative to confocal microscopy in this study. To validate the intracellular distribution of the secretory proteins, observation and image acquisition were also performed using a confocal laser scanning microscope (TCS-SP5; Leica Microsystems), and confocal images of NcGRA6 and NcGRA7 are provided in Appendix A.

### 2.6. In Vivo Infection and Sample Collection

Female mice (*n* = 5) were intraperitoneally injected with 1,000,000 tachyzoites suspended in 400 µL of Roswell Park Memorial Institute (RPMI)-1640 Medium (Sigma-Aldrich). At 30 days post-infection (dpi), all mice were anesthetized, subjected to blood collection by cardiac puncture using a 27-gauge needle and 1 mL syringe, and sacrificed by cervical dislocation. Mouse brains were fixed in 10% formalin neutral buffer solution for histological analysis.

### 2.7. Histopathology and Immunohistochemistry

The formalin-fixed brains were embedded in paraffin, and two-micrometer-thick sections were stained with hematoxylin and eosin (HE) for pathological examination. Immunohistochemistry (IHC) was performed to detect *N. caninum* proteins NcSAG1, NcCYP, NcPF, NcGRA6, and NcGRA7, and heat-induced antigen retrieval was performed using a microwave oven (pH 6). The same rabbit polyclonal antibodies as those used for immunofluorescence were applied during IHC at the same dilution (1:500). The detection system used in the present study was the peroxidase-conjugated EnVision Polymer Detection System (Agilent Technologies, Santa Clara, CA, USA).

### 2.8. Analysis of Staining Patterns

The immunohistochemical staining patterns for NcSAG1, NcCYP, NcPF, NcGRA6, and NcGRA7 were classified as type A when only parasites or both the parasites and the PV space were stained, and as type B when the parasites, the PV space, and the cytoplasm of the infected host cells were stained. To generate binary images, color deconvolution was performed using ImageJ2 (version 2.16.0) to isolate the DAB signal. Thresholding was then applied to the deconvoluted images using a value range of 78–255 to create the final binary images, which were then compared with the original stained images. Parasites without a surrounding DAB signal were classified as type A, whereas those with a detectable signal surrounding them were classified as type B. Classification was based on expert visual assessment by comparing binary and original stained images to distinguish between limited and extended parasite protein diffusion.

### 2.9. Statistical Analysis

The proportions of parasites exhibiting type A and type B staining were evaluated in brain sections from five mice and two calves. For each mouse, more than 30 parasitophorous vacuoles were counted, and for each calf, there were more than 10 parasitophorous vacuoles. All graphs were created using GraphPad Prism (version 10). As the study was primarily descriptive, statistical analyses were not performed.

## 3. Results

### 3.1. Immunofluorescence Revealed the Distribution of N. caninum Proteins in the Infected HFF Cells

We first characterized the distribution of *N. caninum* proteins NcSAG1, NcCYP, NcPF, NcGRA6, and NcGRA7 in the infected HFF cells (Figure 1). NcSAG1, NcCYP, and NcPF were confined to the tachyzoites, with no detectable distribution in the parasitophorous vacuole space or host cell cytoplasm. Conversely, the distribution of NcGRA6 and NcGRA7 was predominantly extracellular, and primarily within the parasitophorous vacuole space and the host cell cytoplasm.

### 3.2. Histological and Immunohistochemical Analysis Showed Pathological Lesions and the Distribution of N. caninum Proteins in the Brains of Experimentally Infected Mice

We characterized our histopathological findings and the distribution of these proteins—NcSAG1, NcCYP, NcPF, NcGRA6, and NcGRA7—in experimentally infected mouse brains. The histological examination revealed multifocal inflammation in the brains of all five mice (Figure 2A). In the lesions, severe infiltration of mononuclear cells, lymphocytes, and macrophages was observed, with prominent perivascular cuffing and meningitis. In addition, multifocal necrosis with calcification and gitter cell infiltration were observed in the cerebral cortex and brain stem of all the infected mice, and numerous tachyzoites were observed in the necrotic and surrounding inflammatory lesions. Immunohistochemistry revealed that NcSAG1, NcCYP, NcPF, NcGRA6, and NcGRA7 were detected in and around the inflammatory and necrotic lesions in the brain of the infected mice, consistent with the distribution of tachyzoites (Figure 2B–F). NcGRA7 was distributed throughout the necrotic lesions in the form of minute granular deposits smaller than the tachyzoites (Figure 2F).

More-detailed observation of the infected mouse brains showed that NcSAG1, NcCYP, and NcPF were only distributed in the tachyzoites (Figure 3A–C), while NcGRA6 and NcGRA7 were also distributed in the neuronal cytoplasm (Figure 3D,E, arrows). The staining patterns revealed that NcSAG1, NcCYP, and NcPF predominantly exhibited type A staining, selectively labeling only the parasites, whereas NcGRA6 and NcGRA7 more frequently showed type B staining, labeling both the parasites and the surrounding host cell cytoplasm (Figure 3F,G and Appendix A). Notably, NcGRA7 exhibited type B staining in nearly all the observed parasitophorous vacuoles, whereas approximately 20% of parasitophorous vacuoles stained for NcGRA6 showed type A staining, restricted to the parasites and the parasitophorous vacuole space.

### 3.3. Histological and Immunohistochemical Analysis Showed Pathological Lesions and the Distribution of N. caninum Proteins in the Brains of Experimentally Infected Calves

Next, we also characterized the histopathological features and protein distribution in the brains of naturally infected calves that exhibited astasia and signs of depression. In the two infected calf brains, as in the mice, multifocal inflammation with severe mononuclear cell infiltration, prominent perivascular cuffing, and meningitis was observed. Although extensive multifocal necrosis with calcification, as observed in the mice, was absent in the calves, prominent glial nodules of varying sizes were present in the cerebral cortex and brainstem. This histological difference suggests distinct host responses between the two species in this study (Figure 4A). Immunohistochemistry revealed that NcCYP, NcPF, NcGRA6, and NcGRA7 (Figure 4C–F) were also detected in and around the inflammatory lesions in the infected calf brains, consistent with the distribution of cysts, and no protozoan organisms were NcSAG1-positive (Figure 4B).

More-detailed observation of the infected calf brains showed that NcSAG1 was absent in the cysts (Figure 5A) and NcCYP was absent in some bradyzoites but strongly expressed in others within the same cyst (Figure 5B). NcPF was distributed in the bradyzoites (Figure 5C), and NcGRA6 and NcGRA7 were distributed in the neuronal cytoplasm in addition to the cysts (Figure 5D,E, arrows). The staining patterns of NcCYP and NcPF in the calves were type A, with NcGRA6 and NcGRA7 being mainly type B (Figure 5F,G and Appendix A).

## 4. Discussion

Immunohistochemistry detected NcSAG1 in almost all the parasites in the infected mouse brains, while NcSAG1 was not detected in the calf brain parasites. NcSAG1 is closely related to *T. gondii* SAG1 and shows tachyzoite stage-specific expression [36,37]. This indicates that the Nc1 strain of *N. caninum* in the mouse brain were in the tachyzoite stage, while those of the *N. caninum* in the calf brain/field were in the bradyzoite stage. This was consistent with previous reports, i.e., tissue cysts have been observed in naturally infected dogs and cattle but not in experimentally infected mice [2,42].

Histopathologically, severe nonsuppurative meningoencephalitis, i.e., perivascular cuffing, multifocal glial nodules, and severe gliosis, was observed in the brains of both the infected mice and calves. However necrotic lesions with gitter cells and calcification were prominently observed in the mice but rarely observed in the calves, and numerous tachyzoites were observed in and around the mouse necrotic lesions. Previous histological examinations of the brains of aborted fetuses infected with *N. caninum* have reported nonsuppurative meningoencephalitis accompanied by necrotic lesions [43,44,45,46]. In addition, tachyzoites were frequently observed in close proximity to these necrotic lesions, whereas tissue cysts were typically located away from the affected areas [45,46]. These findings suggest that tachyzoites are more directly involved in the pathogenesis of necrotizing inflammation in the brain. In the present study, necrotic lesions were not observed in the calves that exhibited emaciation and signs of depression, possibly because reactivation and conversion from cysts to tachyzoites did not occur or was highly restricted in these particular cases, unlike in typical naturally infected cattle exhibiting neurological signs such as limb dysfunction and loss of conscious proprioception [47].

NcGRA6 and NcGRA7 were distributed in the host cytoplasm (Figure 1,Figure 3D,E and Figure 5D,E), with a significantly higher percentage of parasites in the mouse brains showing type B staining patterns, and NcGRA7 exhibited punctate immunoreactivity that was widely distributed throughout necrotic lesions in the infected mice. Previous studies have demonstrated that NcGRA7 localizes to the PVM [27] and can translocate into the host cytoplasm, where it acts in relation to parasite egress, modulates immune responses, and promotes the secretion of inflammatory cytokines, such as IL-6 and IL-12p40, and chemokines, such as CXCL10 and CCL2, in infected mice [34]. In the present study, the detection of NcGRA7 in necrotic lesions suggests that it may be released into the surrounding tissue during host cell death or parasite egress, contributing to inflammation amplification and tissue damage via both direct immune modulation and indirect extracellular signaling.

Immunohistochemistry against secretory proteins in the present study revealed that NcCYP and NcPF antigens were localized in and around the parasites, while NcGRA6 and NcGRA7 antigens were localized in the infected neuronal cytoplasm around the parasites. NcCYP and NcPF have high sequence homology with *T. gondii* cyclophilin 18 and profilin—86% and 98%, respectively—and are involved in host cell invasion and cytokine production regulation [32,48]. These are secretory proteins with signal peptides. However, the present study did not detect them in host cells (Figure 1,Figure 3B,C and Figure 5B,C). This may be attributable to their limited translocation efficiency and low extracellular concentration, resulting in levels insufficient for immunohistochemical detection.

NcGRA6 and NcGRA7 have low sequence homology with *T. gondii* GRA6 and GRA7—28.76% and 29.66%, respectively—and are localized in PV lumen, PVM, and IVN [27]. It can be speculated that NcGRA6 and NcGRA7 were distributed in the host cytoplasm either by translocation into the PVM extensions, by direct translocation into the host cytoplasm, or, as reported for *T. gondii* [12,14,49], by release into the host cytoplasm due to PVM disruption. In *T. gondii*, secreted proteins such as TgGRA7, TgMAF1, TgGRA18, TgGRA16, and TgROP16 are known to be translocated in the PVM (TgGRA7 and TgMAF1), the host cytoplasm (TgGRA18), or nucleus (TgGRA16 and TgROP16), and interact with host organelles and manipulate host metabolism [12,13,14,15,16,17,18,19,20,21]. Recent studies have shown that *N. caninum* also recruits host organelles, salvages lipids [50,51], and alters host metabolic pathways, immune reactions, and cell cycles [52,53,54,55]. While the specific host pathways targeted by NcGRA6 and NcGRA7 remain unclear, their widespread localization in the cytoplasm of infected neurons suggests a possible role in modulating neuronal metabolism, including neural activity.

In this study, brain tissues from five experimentally infected mice and two naturally infected calves were examined. Given that only two naturally infected cases were available, the association between lesion distribution and the localization of parasite-derived proteins remains preliminary; to validate and generalize these findings, future studies should include a larger number of naturally infected animals. Furthermore, we evaluated the intracellular localization of NcGRA6 and NcGRA7 using widefield fluorescence microscopy rather than confocal microscopy. To address this, we employed a Leica Thunder Imaging System with Large Volume Computational Clearing, which effectively suppresses out-of-focus fluorescence and provides enhanced image quality. This method yielded clearer images compared to traditional confocal microscopy, which is known to have higher spatial resolution but can suffer from reduced signal intensity and increased photobleaching in thick sections. Representative confocal images have been included in the Appendix A to support the robustness of our observations.

## 5. Conclusions

In this study, we investigated the distribution of *N. caninum* secretory proteins in the brains of experimentally infected mice and naturally infected calves. NcGRA7 was diffusely detected in necrotic lesions in mice, suggesting its potential role in lesion formation. In contrast, extensive necrosis was not observed in calves, where glial nodules predominated, indicating possible species-specific pathological responses. Moreover, the cytoplasmic localization of NcGRA6 and NcGRA7 within infected host neurons in both species might suggest their involvement in modulating neuronal metabolism, including neuronal activity. These findings highlight species-dependent differences in antigen distribution and lesion development, contributing to a better understanding of *N. caninum* pathogenesis.

## Figures and Tables

**Figure 1 microorganisms-13-01970-f001:**
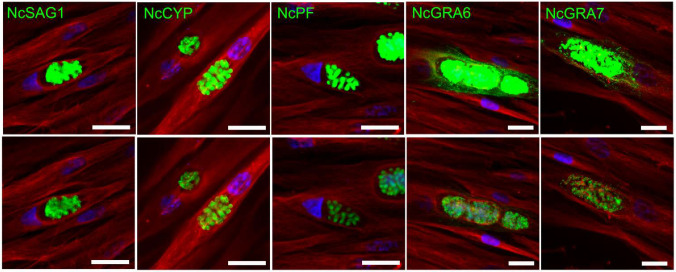
The distribution of *Neospora caninum* proteins, NcSAG1, NcCYP, NcPF, NcGRA6 and NcGRA7, in HFF cells infected with *N. caninum*, analyzed by immunofluorescence. The upper images are the same as the lower ones but with increased green-channel intensity. NcSAG1, NcCYP, and NcPF were localized in the tachyzoites. NcGRA6 and NcGRA7 (green) were localized in the tachyzoites, the parasitophorous vacuole space, and the host cytoplasm. Red: anti-β tubulin; blue: DAPI. Bar = 20 µm.

**Figure 2 microorganisms-13-01970-f002:**
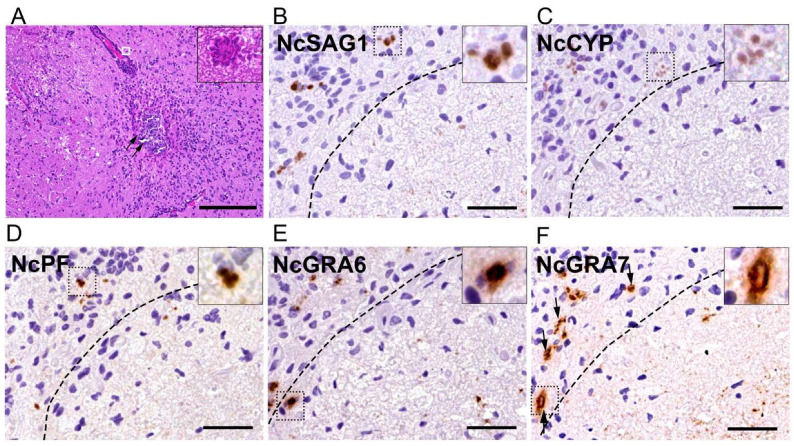
Histopathological and immunohistochemical findings in the brain of mice experimentally infected with *Neospora caninum*. (**A**) A prominent necrotic lesion with calcification (arrows) and gliosis with perivascular cuffing. (Inset of A) A magnified image of the white-box region in A: a group of parasites observed in the mouse brain. Bar = 500 µm. (**B**–**F**) NcSAG1, NcCYP, NcPF, NcGRA6, and NcGRA7 antigens detected in the inflammatory and necrotic lesions of the infected mice. (Insets of **B**–**F**) Magnified images of the dashed-box regions, highlighting the staining patterns of each antigen within the tachyzoites. The NcGRA7 antigen (arrows) was diffusely detected in the necrotic lesions in a fine granular pattern, which differed from the more localized staining observed in the tachyzoites and shown in the insets. Bar = 50 µm.

**Figure 3 microorganisms-13-01970-f003:**
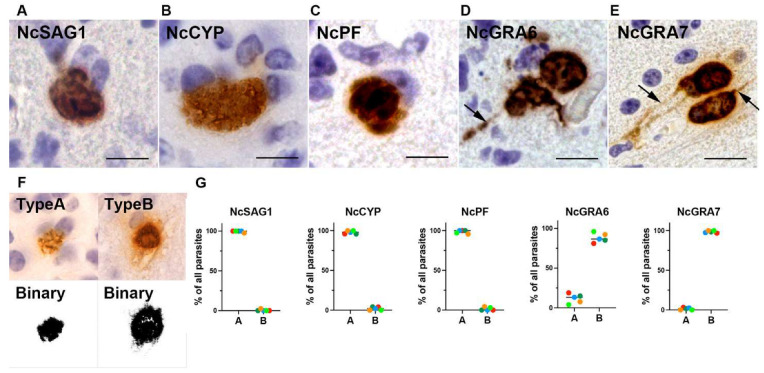
(**A**–**E**) The distribution of *Neospora caninum* proteins in the parasites and host cells of mice experimentally infected with *N. caninum*. NcSAG1, NcCYP, and NcPF were localized in the parasites. NcGRA6 and NcGRA7 were localized in the parasites and the infected neuronal cytoplasm. Bar = 15 µm. (**F**) A classification based on the immunohistochemical staining patterns: type A when only parasites were stained and type B when the parasites, parasitophorous vacuole space, and host cytoplasm were stained. (**G**) The proportions of type A (non-secretory pattern) and type B (secretory pattern) staining as observed in brain sections from five mice experimentally infected with *N. caninum*.

**Figure 4 microorganisms-13-01970-f004:**
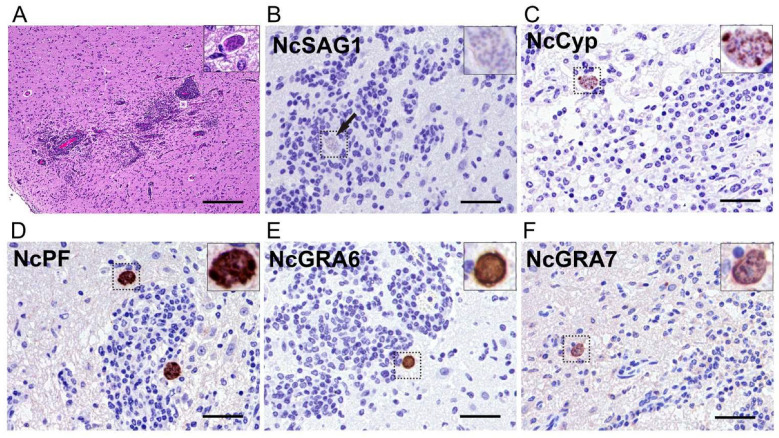
Histopathological and immunohistochemical findings in the brain of calves naturally infected with *Neospora caninum*. (**A**) Gliosis with prominent perivascular cuffing. (Inset of A) A magnified image of the white-box region in A: a group of parasites were observed in the calf. Bar = 500 µm. (**B**) NcSAG1 antigens were not detected in the parasite (arrow). (**C**–**F**) NcCyp, NcPF, NcGRA6, and NcGRA7 antigens were detected in and around the inflammatory lesions of the infected calves. (Insets of B–F) Magnified images of the dashed-box regions, highlighting the staining patterns of each antigen within the cysts. Bar = 50 µm.

**Figure 5 microorganisms-13-01970-f005:**
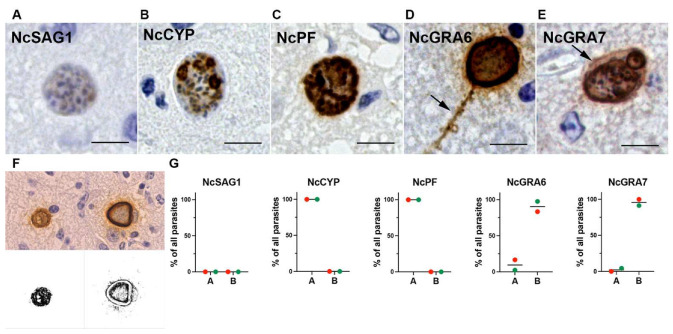
(**A**–**E**) The distribution of *Neospora caninum* proteins in the parasites and host cells of calves naturally infected with *N. caninum*. Few NcSAG1 antigens were detected in the parasites; NcCYP and NcPF were localized in the parasites within the cyst; and NcGRA6 and NcGRA7 were localized in the parasites and the infected neuronal cytoplasm. Bar = 15 µm. (**F**) A classification based on the immunohistochemical staining patterns: type A when only parasites were stained and type B when the parasites, parasitophorous vacuole space, and host cytoplasm were stained. (**G**) The proportions of type A (non-secretory pattern) and type B (secretory pattern) were observed in brain sections from the two calves naturally infected with *N. caninum*.

## Data Availability

The original contributions presented in the study are included in the article/Appendix A; further inquiries can be directed to the corresponding author.

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
