# Peer review of "The Distribution of Neospora caninum Secretory Proteins in Mouse and Calf Brains"

_microorganisms, 2025, doi:10.3390/microorganisms13091970_

Round 1
Reviewer 1 Report
Comments and Suggestions for Authors
Here, we described the brain distribution and the intracellular distribution of the secretory proteins in experimentally infected mice and naturally infected calves by histopathology, immunohistochemistry (IHC) for surface antigen 1 (NcSAG1), cyclophilin (NcCYP), profilin (NcPF), dense granule protein 6 (NcGRA6), and NcGRA7
The study reproduces the action of proteins and their distribution; however, to increase reliability, it would be advisable to expand the sample (Materials and Methods).
The conclusion can be substantially improved. Consider aligning it with the species and distribution
Consider updating the references, especially in the introduction section.

Author Response
Author's Notes to Reviewer 1
Here, we described the brain distribution and the intracellular distribution of the secretory proteins in experimentally infected mice and naturally infected calves by histopathology, immunohistochemistry (IHC) for surface antigen 1 (NcSAG1), cyclophilin (NcCYP), profilin (NcPF), dense granule protein 6 (NcGRA6), and NcGRA7
Thank you very much for reviewing our manuscript (microorganisms-3777056). Your insightful comments have helped us improve our paper. We are here submitting our revised manuscript entitled “The Distribution of Neospora caninum Secretory Proteins in the Brain of Mice and Calves” for your kind consideration of its suitability for publication in Microorganisms. Our incorporation of the reviewers’ suggestions are as follows.
Comment 1: The study reproduces the action of proteins and their distribution; however, to increase reliability, it would be advisable to expand the sample (Materials and Methods).
Response: Thank you for consideration and helpful comments on our manuscript. At present, only two neonatal calf samples are available for this study, and acquiring additional specimens is currently unfeasible due to practical limitations. In line with the reviewer’s valuable comment, we acknowledge that a larger sample size would enhance the reliability and reproducibility of the findings. Accordingly, this limitation has been explicitly addressed in the Discussion section of the revised manuscript.
>Line 479-483 (Discussion section)
In this study, brain tissues from five experimentally infected mice and two naturally infected calves were examined. Given that only two naturally infected cases were available, the association between lesion distribution and the localization of parasite-derived proteins remains preliminary. To validate and generalize these findings, future studies should include a larger number of naturally infected animals.
Comment 2: The conclusion can be substantially improved. Consider aligning it with the species and distribution.
Response: Thank you for consideration and helpful comments on our manuscript. As your comment, we have revised the description of the conclusion.
>Line 494-503
In this study, we investigated the distribution of N. caninum secretory proteins in the brains of experimentally infected mice and naturally infected calves. NcGRA7 was diffusely detected in necrotic lesions in mice, suggesting its potential role in lesion formation. In contrast, extensive necrosis was not observed in calves, where glial nodules predominated, indicating possible species-specific pathological responses. Moreover, the cytoplasmic localization of NcGRA6 and NcGRA7 within infected host neurons in both species suggests their involvement in modulating neuronal metabolism, including neuronal activity. These findings highlight species-dependent differences in antigen distribution and lesion development, contributing to a better understanding of N. caninumpathogenesis.
Comment 3: Consider updating the references, especially in the introduction section.
Response: Thank you for consideration and helpful comments on our manuscript. As your comment, we have carefully reviewed and updated the references cited in the introduction section.
> Line 33-34.
It causes substantial economic losses in the beef and dairy industries [6,7].
- Parraguez M.C.; Ponssa E.; Caffarena D.; Artagaveytia J.; Sotelo F.; Fariña S.; Mendoza A.; Giannitti F.. Estimation of direct economic and productive losses due to abortions caused by Neospora caninum in the primary dairy sector of Uruguay. Front Vet Sci. 2025, 26, 12, 1502742. https://doi: 10.3389/fvets.2025.1502742.
- Reichel M.P.; Ayanegui-Alcérreca M.A.; Gondim L.F.P.; Ellis J.T.. What is the global economic impact of Neospora caninum in cattle - the billion dollar question. Int J Parasitol. 2013, 43(2), 133-42. https://doi: 10.1016/j.ijpara.2012.10.022.
> Line 66-68.
NcSAG1 is abundantly expressed on the plasma membrane of tachyzoites and widely studied for diagnostic marker [38-40], making it a useful reference for comparison with secretory proteins.
- Dellarupe A.; Moré G.; Unzaga J.M.; Pardini L.; Venturini M.C.. Study of specific immunodominant antigens in different stages of Neospora caninum, Toxoplasma gondii, Sarcocystis spp. and Hammondia Exp Parasitol. 2024, 262, 108772. https://doi: 10.1016/j.exppara.2024.108772.
- Udonsom R.;, Adisakwattana P.; Popruk S.;, Reamtong O.; Jirapattharasate C.; Thiangtrongjit T.; Rerkyusuke S.; Chanlun A.; Hasan T.;, Kotepui M.; et al. Evaluation of Immunodiagnostic Performances of Neospora caninum Peroxiredoxin 2 (NcPrx2), Microneme 4 (NcMIC4), and Surface Antigen 1 (NcSAG1) Recombinant Proteins for Bovine Neosporosis. Animals (Basel). 2024, 4(4), 531. https://doi: 10.3390/ani14040531.
Response: Thank you for consideration and helpful comments on our manuscript. As your comment, we have revised the description.
>Line 355-359.
Immunohistochemistry revealed that NcCYP, NcPF, NcGRA6, and NcGRA7 (Figures 4C-F) were also detected in and around the inflammatory lesions in the brain of the infected calves, which were consistent with the distribution of cysts and no protozoan organisms exhibited positive for NcSAG1 (Figure 4 B).
>Line 414.
NcGRA6 and NcGRA7 were distributed in the host cytoplasm (Figures 1, 3D 3E, 5D, and 5E).
>Line 433.
These proteins have signal peptides and are secretory proteins. However, the present study did not detect them in the host cell (Figures 1, 3B, 3C, 5B, and 5C).
>Line 494-503.
In this study, we investigated the distribution of N. caninum secretory proteins in the brains of experimentally infected mice and naturally infected calves. NcGRA7 was diffusely detected in necrotic lesions in mice, suggesting its potential role in lesion formation. In contrast, extensive necrosis was not observed in calves, where glial nodules predominated, indicating possible species-specific pathological responses. Moreover, the cytoplasmic localization of NcGRA6 and NcGRA7 within infected host neurons in both species might suggest their involvement in modulating neuronal metabolism, including neuronal activity. These findings highlight species-dependent differences in antigen distribution and lesion development, contributing to a better understanding of N. caninumpathogenesis.

Reviewer 2 Report
Comments and Suggestions for Authors
The manuscript submitted by Usio-Watanabe et al. aims to characterize cellular distribution of Neospora caninum-derived secretory proteins at cell culture and histological levels, and to link its distribution with inflammatory responses observed in N. caninum-infected mice and calves. The topic is relevant, and the findings are of interest to the field of host-parasite interactions. However, substantial revisions are required before the manuscript can be considered for publication.
Since line numbers are not provided, comments are organized by section and paragraph.
Introduction: No comments
Material and methods:
The description of the production and purification of the polyclonal antibodies is absent. If this information is referred to piously published works, it is recommended that this information be integrated with the subsequent immunofluorescence paragraph. Furthermore, the dilution factor of each antibody used in the different techniques should be explicitly stated. Additionally, in section 2.7, please refer to the antibodies described earlier to improve clarity.
Section 2.8 requires substantial improvement. As one of the core aspects of the study, the image analysis methodology is insufficiently described. Please include the name and version of the software used for image analysis, all image adjustment procedures, and the thresholding techniques employed to generate binary images. Additionally, the classification of staining patterns into types A and B must be clarified—if this classification is based on prior literature, proper references must be cited; if it is a novel contribution, that should be clearly stated.
The statistical analysis section is absent. Even if no statistical test is performed, the software used to generate the presented graphs must be indicated.
Results:
Immunofluorescence: The use of epifluorescence microscopy appears to be insufficient for the claims made regarding extra parasitophorous vacuole signal localization, particularly for markers such as GRA6 and GRA7. Although the authors attempt to address this by modifying channel intensity (which is not documented in the Methods section), this approach does not adequately support the conclusion. It is strongly recommended to include confocal microscopy data to provide higher-resolution evidence. This limitation should also be discussed explicitly in the Discussion section.
Histological analysis in infected animals: Given the limited sample size (mice: n=5; calves: n=2), it is essential to include a quantitative summary of the total number of parasitophorous vacuoles analysed, along with the staining patterns for each marker (SAG1, CYP, PF, GRA6, GRA7). This will help demonstrate the reproducibility and consistency of the immunostaining results despite the limited number of animals.
Discussion: No comments
Author Response
Author's Notes to Reviewer 2
The manuscript submitted by Usio-Watanabe et al. aims to characterize cellular distribution of Neospora caninum-derived secretory proteins at cell culture and histological levels, and to link its distribution with inflammatory responses observed in N. caninum-infected mice and calves. The topic is relevant, and the findings are of interest to the field of host-parasite interactions. However, substantial revisions are required before the manuscript can be considered for publication.
Thank you very much for reviewing our manuscript (microorganisms-3777056). Your insightful comments have helped us improve our paper. We are here submitting our revised manuscript entitled “The Distribution of Neospora caninumSecretory Proteins in the Brain of Mice and Calves” for your kind consideration of its suitability for publication in Microorganisms. Our incorporation of the reviewers’ suggestions are as follows.
Since line numbers are not provided, comments are organized by section and paragraph.
Introduction: No comments
Material and methods:
Comment 1: The description of the production and purification of the polyclonal antibodies is absent. If this information is referred to piously published works, it is recommended that this information be integrated with the subsequent immunofluorescence paragraph. Furthermore, the dilution factor of each antibody used in the different techniques should be explicitly stated. Additionally, in section 2.7, please refer to the antibodies described earlier to improve clarity.
Response:  Thank you for consideration and helpful comments on our manuscript. As your comment, we have revised the description of the materials and methods in sections 2.4, 2.5 and 2.7.
>Line 196-205. (2.4. Production of polyclonal antibodies against N. caninum.)
Briefly, NcSAG1 gene (nt 89-896), NcCYP gene (nt 54-537), NcPF gene (nt 1-492), NcGRA6 gene (nt 130–462) and NcGRA7 gene (nt 1-711) were amplified with PCR from cDNA of N. caninum tachyzoites and cloned into the pGEX4T-1 plasmid. The recombinant proteins were expressed as glutathione S-transferase (GST) fusion proteins in Escherichia coli strain DH5a (Takara Bio Inc., Shiga, Japan). Each recombinant protein (300μg) in Freund’s complete adjuvant (Sigma) was injected intradermally into a female Japanese white rabbit (Kitayama Labes, Nagano, Japan) on day 0, 14, 28, and 42. IgG was purified from 2 ml of serum collected 7 days after the last immunization using a protein A chromatography column (Econo-Pac® Protein A Kit, Bio-Rad Laboratories, CA, USA), according to the manufacturer’s instructions.
>Line 218. (2.5. Immunofluorescence)
After blocking with 3% bovine serum albumin (BSA, FUJIFILM), samples were incubated with rabbit polyclonal antibodies against NcSAG1, NcCYP, NcPF, NcGRA6 and NcGRA7 (all diluted 1:500) for 1 hour at 37°C.
>Line 258-259. (2.7. Histopathology and immunohistochemistry)
The same rabbit polyclonal antibodies as those used for immunofluorescence were applied in IHC at the same dilution (1:500).
Comment 2: Section 2.8 requires substantial improvement. As one of the core aspects of the study, the image analysis methodology is insufficiently described. Please include the name and version of the software used for image analysis, all image adjustment procedures, and the thresholding techniques employed to generate binary images. Additionally, the classification of staining patterns into types A and B must be clarified—if this classification is based on prior literature, proper references must be cited; if it is a novel contribution, that should be clearly stated.
Response:  Thank you for consideration and helpful comments on our manuscript. As your comment, we have revised the description of the materials and methods.
> Line 266-273.
To generate binary images, color deconvolution was performed using ImageJ2 (version 2.16.0) to isolate the DAB signal. Thresholding was then applied to the deconvoluted image using a value range of 78–255 to create the final binary image. These binary images were compared with the original stained images. Parasites without a surrounding DAB signal were classified as type A, whereas those with a detectable signal surrounding them were classified as type B. Classification was based on expert visual assessment by comparing binary and original stained images, to distinguish between limited and extended diffusion of parasite proteins.
Comment 3: The statistical analysis section is absent. Even if no statistical test is performed, the software used to generate the presented graphs must be indicated.
Response:  Thank you for consideration and helpful comments on our manuscript. As your comment, we have revised the description of the materials and methods.
> Line 274-279.
2.9. Statistical analysis
The proportions of parasites exhibiting type A and type B staining were evaluated in brain sections from five mice and two calves. For each mouse, more than 30 para-sitophorous vacuoles were counted. For each calf, more than 10 parasitophorous vacuoles were counted. All graphs were created using GraphPad Prism (version 10). As the study was primarily descriptive, statistical analyses were not performed.
Results:
Comment 4: Immunofluorescence: The use of epifluorescence microscopy appears to be insufficient for the claims made regarding extra parasitophorous vacuole signal localization, particularly for markers such as GRA6 and GRA7. Although the authors attempt to address this by modifying channel intensity (which is not documented in the Methods section), this approach does not adequately support the conclusion. It is strongly recommended to include confocal microscopy data to provide higher-resolution evidence. This limitation should also be discussed explicitly in the Discussion section.
Response: Thank you for consideration and helpful comments on our manuscript. We performed confocal microscopy to assess the intracellular localization of NcGRA6 and NcGRA7. However, the resulting images had lower clarity compared to those obtained using the Leica Thunder system. Therefore, we opted to use the images acquired with the Leica Thunder microscope in the main figures. In response to the reviewer’s comment, we will include representative confocal images as supplementary data. In this study, we used the Large Volume Computational Clearing (LVCC) system of the Leica Thunder microscope. This system effectively removes out-of-focus fluorescence signals typically associated with epifluorescence microscopy. As such, it was used in place of traditional confocal microscopy for high-resolution image acquisition. We will include this methodological detail in the revised Materials and Methods section and the limitation of LVCC in Discussion section.
> Line 226-245. (Materials and methods)
Observation and image acquisition were performed using the Leica Thunder Imaging System equipped with the Large Volume Computational Clearing (LVCC) module (Leica Microsystems GmbH, Wetzlar, Germany). This system is based on widefield epifluorescence microscopy and employs computational algorithms to eliminate out-of-focus light, thereby enhancing image clarity and contrast. The LVCC method allows for the visualization of intracellular protein localization with improved resolution and minimal photobleaching, and was used as an alternative to confocal microscopy in this study. Confocal microscopy was also performed for validation, and confocal images of NcGRA6 and NcGRA7 are provided in Figure S1.
>Line 483-492. (Discussion)
Furthermore, in the study we evaluated the intracellular localization of NcGRA6 and NcGRA7 by using widefield fluorescence microscopy rather than confocal microscopy. To address this, we employed the Leica Thunder Imaging System with Large Volume Computational Clearing, which effectively suppresses out-of-focus fluorescence and provides enhanced image quality. In our hands, this method yielded clearer images compared to traditional confocal microscopy, which is known to have higher spatial resolution but can suffer from reduced signal intensity and increased photobleaching in thick sections. Representative confocal images have been included in the supplementary materials to support the robustness of our observations.
Figure S1 legend: Confocal microscopy images of NcGRA6 and NcGRA7(green). Green signals are observed in the cytoplasm of infected cells. Red: anti-β tubulin. Bar= 20 µm.
>Line 505-507.
Supplementary Materials: The following supporting information can be downloaded at: https://www.mdpi.com/article/doi/s1, Figure S1: Confocal microscopy images of NcGRA6 and NcGRA7;
Comment 5: Histological analysis in infected animals: Given the limited sample size (mice: n=5; calves: n=2), it is essential to include a quantitative summary of the total number of parasitophorous vacuoles analysed, along with the staining patterns for each marker (SAG1, CYP, PF, GRA6, GRA7). This will help demonstrate the reproducibility and consistency of the immunostaining results despite the limited number of animals.
Response: Thank you for consideration and helpful comments on our manuscript. As your comment, we have provided Table S1 and add the description of the result.
> Line 327.
The staining patterns revealed that NcSAG1, NcCYP, and NcPF predominantly exhibited type A staining, selectively labeling only the parasites, whereas NcGRA6 and NcGRA7 more frequently showed type B staining, labeling both the parasites and the surrounding host cell cytoplasm (Figures 3 F, G, and Table S1).
>Line 379.
Staining patterns of NcCYP and NcPF in the calves were type A, and that of NcGRA6 and NcGRA7 were mainly type B (Figures 5 F, G and Table S1).
>Line 505-508.

Reviewer 3 Report
Comments and Suggestions for Authors
The manuscript entitled “The Distribution of Neospora caninum Secretory Proteins in the Brain of Mice and Calves” presents an investigation into the spatial distribution and intracellular localization of Neospora caninum secretory proteins in experimentally infected mice and naturally infected calves. The authors employed histopathology and immunohistochemistry to evaluate the presence of five antigens: NcSAG1, NcCYP, NcPF, NcGRA6, and NcGRA7.
The topic is scientifically relevant and aligns well with the scope of Microorganisms. It advances the understanding of parasite-host interactions, especially the tissue and subcellular localization of important secretory proteins. However, several methodological and interpretive issues need clarification or strengthening to improve the manuscript’s scientific rigor and ensure the validity of its conclusions.
Major Comments:
Introduction:
• Please add a citation to support the statement: “It causes substantial economic losses in the beef and dairy industries.”
• The relevance of the study is appropriately introduced; however, the novelty would be better highlighted by noting that current data on secretory protein distribution are limited to murine models (e.g., reference 33). It should be noted that data on NcCYP, NcPF, NcGRA6, and even NcGRA7 are currently lacking in cattle.
• The inclusion of the surface protein NcSAG1 alongside secretory proteins requires further justification. A sentence describing its biological role and diagnostic value in Neospora caninum would help contextualize its relevance.
Materials and Methods:
• Section 2.8 “Analysis of the staining pattern” should describe the complete set of criteria used to classify immunohistochemical patterns as type A or B. This would enhance reproducibility and interpretation of the results.
Results:
• Figure 1 legend: Please add “by immunofluorescence” after “HFF cells” for clarity.
• The methodology used in Figure 1 is insufficient to determine subcellular localization with certainty in the absence of co-staining with organelle-specific markers or the use of confocal microscopy. This limitation, along with others, should be acknowledged in the discussion.
• Figure 2: Clarify what the arrows indicate in panels A and F. The statement “which was not consistent with the distribution of tachyzoites” is vague and should be clarified, especially for readers unfamiliar with the typical localization of tachyzoites. Consider improving image resolution and updating the figure legend to be self-explanatory.
• Figure 3 (panels F and G): Specify whether the classification of staining patterns into types A and B was based on quantitative pixel analysis or expert visual assessment.
• Figure 4: To improve visual interpretation, an inset like the one in panel A should be included for all subsets.
Discussion:
• Some histopathological observations reported in the discussion (e.g., differences in necrotic lesions between mice and calves) are not described in the results section. These should either be integrated into the results or omitted from the discussion.
• The discussion would benefit from a more thorough evaluation of methodological limitations and a cautious interpretation of the findings, particularly regarding protein localization and lesion severity.
The manuscript presents novel findings, particularly in the context of Neospora protein distribution in calves, a host less commonly explored. However, its interpretability and scientific impact would be significantly improved through the inclusion of semi-quantitative data and a more detailed methodological description.
Author Response
Author's Notes to Reviewer 3
The manuscript entitled “The Distribution of Neospora caninum Secretory Proteins in the Brain of Mice and Calves” presents an investigation into the spatial distribution and intracellular localization of Neospora caninum secretory proteins in experimentally infected mice and naturally infected calves. The authors employed histopathology and immunohistochemistry to evaluate the presence of five antigens: NcSAG1, NcCYP, NcPF, NcGRA6, and NcGRA7.
The topic is scientifically relevant and aligns well with the scope of Microorganisms. It advances the understanding of parasite-host interactions, especially the tissue and subcellular localization of important secretory proteins. However, several methodological and interpretive issues need clarification or strengthening to improve the manuscript’s scientific rigor and ensure the validity of its conclusions.
Thank you very much for reviewing our manuscript (microorganisms-3777056). Your insightful comments have helped us improve our paper. We are here submitting our revised manuscript entitled “The Distribution of Neospora caninum Secretory Proteins in the Brain of Mice and Calves” for your kind consideration of its suitability for publication in Microorganisms. Our incorporation of the reviewers’ suggestions are as follows.
Major Comments:
Introduction:
Comment 1: Please add a citation to support the statement: “It causes substantial economic losses in the beef and dairy industries.”
Response: Thank you for consideration and helpful comments on our manuscript. As your comment, we have added the following references.
>Line 34.
- Parraguez M.C.; Ponssa E.; Caffarena D.; Artagaveytia J.; Sotelo F.; Fariña S.; Mendoza A.; Giannitti F.. Estimation of direct economic and productive losses due to abortions caused by Neospora caninum in the primary dairy sector of Uruguay. Front Vet Sci. 2025, 26, 12, 1502742. https://doi: 10.3389/fvets.2025.1502742.
- Reichel M.P.; Ayanegui-Alcérreca M.A.; Gondim L.F.P.; Ellis J.T.. What is the global economic impact of Neospora caninum in cattle - the billion dollar question. Int J Parasitol. 2013, 43(2), 133-42. https://doi: 10.1016/j.ijpara.2012.10.022.
Comment 2: The relevance of the study is appropriately introduced; however, the novelty would be better highlighted by noting that current data on secretory protein distribution are limited to murine models (e.g., reference 33). It should be noted that data on NcCYP, NcPF, NcGRA6, and even NcGRA7 are currently lacking in cattle.
Response: Thank you for consideration and helpful comments on our manuscript. As your comment, we have revised the introduction.
> Line 62-70.
Notably, even for NcGRA7, no study has examined the relationship between the distributions of these secretory proteins and lesions in cattle. In the present study, we focused on the distribution of NcCYP, NcPF, NcGRA6, and NcGRA7, in addition to the non-secretory protein, N. caninum surface antigen 1 (NcSAG1) [37,38], in the brains of experimentally infected mice and naturally infected calves. NcSAG1 is abundantly expressed on the plasma membrane of tachyzoites and widely studied for diagnostic marker [38-40], making it a useful reference for comparison with secretory proteins. This highlights a significant gap in our understanding of the pathogenesis of N. caninum infection in its natural hosts.
Comment 3: The inclusion of the surface protein NcSAG1 alongside secretory proteins requires further justification. A sentence describing its biological role and diagnostic value in Neospora caninum would help contextualize its relevance.
Response: Thank you for consideration and helpful comments on our manuscript. As your comment, we have revised the introduction.
> Line 66-70.
NcSAG1 is abundantly expressed on the plasma membrane of tachyzoites and widely studied for diagnostic marker [38-40], making it a useful reference for comparison with secretory proteins.
- Dellarupe A.; Moré G.; Unzaga J.M.; Pardini L.; Venturini M.C.. Study of specific immunodominant antigens in different stages of Neospora caninum, Toxoplasma gondii, Sarcocystis spp. and Hammondia Exp Parasitol. 2024, 262, 108772. https://doi: 10.1016/j.exppara.2024.108772.
- Udonsom R.;, Adisakwattana P.; Popruk S.;, Reamtong O.; Jirapattharasate C.; Thiangtrongjit T.; Rerkyusuke S.; Chanlun A.; Hasan T.;, Kotepui M.; et al. Evaluation of Immunodiagnostic Performances of Neospora caninum Peroxiredoxin 2 (NcPrx2), Microneme 4 (NcMIC4), and Surface Antigen 1 (NcSAG1) Recombinant Proteins for Bovine Neosporosis. Animals (Basel). 2024, 4(4), 531. https://doi: 10.3390/ani14040531.
Materials and Methods:
Comment 4: Section 2.8 “Analysis of the staining pattern” should describe the complete set of criteria used to classify immunohistochemical patterns as type A or B. This would enhance reproducibility and interpretation of the results.
Response:  Thank you for consideration and helpful comments on our manuscript. As your comment, we have revised the description of the materials and methods.
> Line 266-273.
To generate binary images, color deconvolution was performed using ImageJ2 (version 2.16.0) to isolate the DAB signal. Thresholding was then applied to the deconvoluted image using a value range of 78–255 to create the final binary image. These binary images were compared with the original stained images. Parasites without a surrounding DAB signal were classified as type A, whereas those with a detectable signal surrounding them were classified as type B. Classification was based on expert visual assessment by comparing binary and original stained images, to distinguish between limited and extended diffusion of parasite proteins.
Results:
Comment 5: Figure 1 legend: Please add “by immunofluorescence” after “HFF cells” for clarity.
Response:  Thank you for consideration and helpful comments on our manuscript. As your comment, we have revised the description in Figure 1 legend.
> Line 291.
Figure 1. The distribution of N. caninum proteins in HFF cells by immunofluorescence.
Comment 6: The methodology used in Figure 1 is insufficient to determine subcellular localization with certainty in the absence of co-staining with organelle-specific markers or the use of confocal microscopy. This limitation, along with others, should be acknowledged in the discussion.
Response: Thank you for consideration and helpful comments on our manuscript. We performed confocal microscopy to assess the intracellular localization of NcGRA6 and NcGRA7. However, the resulting images had lower clarity compared to those obtained using the Leica Thunder system. Therefore, we opted to use the images acquired with the Leica Thunder microscope in the main figures. In response to the reviewer’s comment, we will include representative confocal images as supplementary data. In this study, we used the Large Volume Computational Clearing (LVCC) system of the Leica Thunder microscope. This system effectively removes out-of-focus fluorescence signals typically associated with epifluorescence microscopy. As such, it was used in place of traditional confocal microscopy for high-resolution image acquisition. We will include this methodological detail in the revised Materials and Methods section and the limitation of LVCC in Discussion section.
> Line 226-245. (Materials and methods)
Observation and image acquisition were performed using the Leica Thunder Imaging System equipped with the Large Volume Computational Clearing (LVCC) module (Leica Microsystems GmbH, Wetzlar, Germany). This system is based on widefield epifluorescence microscopy and employs computational algorithms to eliminate out-of-focus light, thereby enhancing image clarity and contrast. The LVCC method allows for the visualization of intracellular protein localization with improved resolution and minimal photobleaching, and was used as an alternative to confocal microscopy in this study. Confocal microscopy was also performed for validation, and confocal images of NcGRA6 and NcGRA7 are provided in Figure S1.
>Line 483-492. (Discussion)
Furthermore, in the study we evaluated the intracellular localization of NcGRA6 and NcGRA7 by using widefield fluorescence microscopy rather than confocal microscopy. To address this, we employed the Leica Thunder Imaging System with Large Volume Computational Clearing, which effectively suppresses out-of-focus fluorescence and provides enhanced image quality. In our hands, this method yielded clearer images compared to traditional confocal microscopy, which is known to have higher spatial resolution but can suffer from reduced signal intensity and increased photobleaching in thick sections. Representative confocal images have been included in the supplementary materials to support the robustness of our observations.
Figure S1 legend: Confocal microscopy images of NcGRA6 and NcGRA7. NcGRA6 and NcGRA7 (green signals) are observed in the cytoplasm of infected cells. Red: anti-β tubulin. Bar= 20 µm.
>Line 505-507.
Supplementary Materials: The following supporting information can be downloaded at: https://www.mdpi.com/article/doi/s1, Figure S1: Confocal microscopy images of NcGRA6 and NcGRA7;
Comment 7: Figure 2: Clarify what the arrows indicate in panels A and F. The statement “which was not consistent with the distribution of tachyzoites” is vague and should be clarified, especially for readers unfamiliar with the typical localization of tachyzoites. Consider improving image resolution and updating the figure legend to be self-explanatory.
Response:  Thank you for consideration and helpful comments on our manuscript. As your comment, we have revised Figure2 and legend. In Figure 2, we aimed to provide an overview of the spatial relationship between the distribution of necrotic lesions and tachyzoites. Therefore, we limited the magnification of the main image to preserve the overall context, while using an inset to highlight the staining characteristics of the tachyzoites. We believe this approach effectively conveys to the reader the difference between the specific staining of tachyzoites and the diffuse staining of NcGRA7 within the necrotic lesions.
>Line 313-319. (Figure 2 legend)
Histopathological and immunohistochemical findings in the brain of the infected mouse. (A) Prominent necrotic lesion with calcification (arrows) and gliosis with perivascular cuffing. (Inset of A) Magnified image of the white box region in A. A group of parasites were observed in the mice. Bar=500 µm. (B-F) NcSAG1, NcCYP, NcPF, NcGRA6, and NcGRA7 antigens were detected in the inflammatory and necrotic lesions of the infected mice. (Inset of B-F) Magnified images of the dashed-box regions, highlighting the staining patterns of each antigen within the tachyzoites. NcGRA7 antigen was diffusely detected in the necrotic lesions in a fine granular pattern, which differed from the more localized staining observed in tachyzoites shown in the insets. Bar= 50 µm
Comment 8: Figure 3 (panels F and G): Specify whether the classification of staining patterns into types A and B was based on quantitative pixel analysis or expert visual assessment.
Response:  Thank you for consideration and helpful comments on our manuscript. As your comment, we have revised the description of materials and methods.
> Line 271-273.
Classification was based on expert visual assessment by comparing binary and original stained images, to distinguish between limited and extended diffusion of parasite proteins.
Comment 9: Figure 4: To improve visual interpretation, an inset like the one in panel A should be included for all subsets.
Response:  Thank you for consideration and helpful comments on our manuscript. As your comment, we have revised figure4 and figure 4 legend.
> Line 366-371. (Figure 4 legend)
Histopathological and immunohistochemical findings in the brain of the infected calf. (A) Gliosis with prominent perivascular cuffing. (Inset of A) Magnified image of the white box region in A. A group of parasites were observed in the calf. Bar= 500 µm. (B) NcSAG1 antigens were not detected in the parasite (arrow). (C-F) NcCyp, NcPF, NcGRA6 and NcGRA7 antigens were detected in and around the inflammatory lesions of the infected calves. (Inset of B-F) Magnified images of the dashed-box regions, highlighting the staining patterns of each antigen within the cyst. Bar= 50 µm.
Discussion:
Comment 10: Some histopathological observations reported in the discussion (e.g., differences in necrotic lesions between mice and calves) are not described in the results section. These should either be integrated into the results or omitted from the discussion.
Response: Thank you for consideration and helpful comments on our manuscript. To address this, we revised the Results section to clearly state the histopathological differences between mice and calves, especially regarding the presence or absence of necrotic lesions. This ensures consistency between the Results and Discussion sections.
> Line 354-355.
In the brain of the infected two calves, as in the mice, multifocal inflammation with severe mononuclear cells infiltration, prominent perivascular cuffing and meningitis were observed. Although extensive multifocal necrosis with calcification, as observed in the mice, was absent in the calves, prominent glial nodules of varying sizes were present in the cerebral cortex and brainstem. This histological difference suggests a distinct host response between the two species in this study (Figure 4 A).
Comment 11: The discussion would benefit from a more thorough evaluation of methodological limitations and a cautious interpretation of the findings, particularly regarding protein localization and lesion severity.
Response: Thank you for consideration and helpful comments on our manuscript. As your comment, we have added the description on the Discussion.
>Line 479-492.
In this study, brain tissues from five experimentally infected mice and two naturally infected calves were examined. Given that only two naturally infected cases were available, the association between lesion distribution and the localization of parasite-derived proteins remains preliminary. To validate and generalize these findings, future studies should include a larger number of naturally infected animals. Furthermore, in the study we evaluated the intracellular localization of NcGRA6 and NcGRA7 by using widefield fluorescence microscopy rather than confocal microscopy. To address this, we employed the Leica Thunder Imaging System with Large Volume Computational Clearing, which effectively suppresses out-of-focus fluorescence and provides enhanced image quality. In our hands, this method yielded clearer images compared to traditional confocal microscopy, which is known to have higher spatial resolution but can suffer from reduced signal intensity and increased photobleaching in thick sections. Representative confocal images have been included in the supplementary materials to support the robustness of our observations.
Comment 12: The manuscript presents novel findings, particularly in the context of Neospora protein distribution in calves, a host less commonly explored. However, its interpretability and scientific impact would be significantly improved through the inclusion of semi-quantitative data and a more detailed methodological description.
Response: Thank you for your comment regarding the need for semi-quantitative analysis and methodological clarity. We acknowledge the limitation of a small sample size for the calf group, which restricts robust quantitative analysis. However, to strengthen the reliability of our interpretation, we have provided the detailed staining results for each marker in a supplementary table. Furthermore, we have expanded the Materials and Methods section to include a more detailed description of how staining patterns were evaluated. We believe that these additions address the reviewer’s concerns regarding both data interpretability and methodological clarity.
>Line 266-273.
To generate binary images, color deconvolution was performed using ImageJ2 (version 2.16.0) to isolate the DAB signal. Thresholding was then applied to the deconvoluted image using a value range of 78–255 to create the final binary image. These binary images were compared with the original stained images. Parasites without a surrounding DAB signal were classified as type A, whereas those with a detectable signal surrounding them were classified as type B. Classification was based on expert visual assessment by comparing binary and original stained images, to distinguish between limited and extended diffusion of parasite proteins.
>Line 324-327.
The staining patterns revealed that NcSAG1, NcCYP, and NcPF predominantly exhibited type A staining, selectively labeling only the parasites, whereas NcGRA6 and NcGRA7 more frequently showed type B staining, labeling both the parasites and the surrounding host cell cytoplasm (Figures 3 F, G, and Table S1).
>Line 378-379.
Staining patterns of NcCYP and NcPF in the calves were type A, and that of NcGRA6 and NcGRA7 were mainly type B (Figures 5 F, G and Table S1).
>Line 505-508.

Round 2
Reviewer 2 Report
Comments and Suggestions for Authors
The authors included most of the comments.
Author Response
Comments and Suggestions for Authors
The authors included most of the comments.
Response: Thank you very much for reviewing our revised manuscript (microorganisms-3777056). Your insightful comments have helped us improve our paper. To address the reviewer’s comments on the English, we requested an English editing service from MDPI Author Services and added this information to Acknowledgments section. Revisions made by the editing service are shown in blue, and further revisions made in response to the editor’s suggestions are shown in red. We are here submitting our revised manuscript entitled “The Distribution of Neospora caninum Secretory Proteins in Mouse and Calf Brains” for your kind consideration of its suitability for publication in Microorganisms. Our incorporation of the editor’s and reviewers’ suggestions are as follows.

Reviewer 3 Report
Comments and Suggestions for Authors
The authors have incorporated the suggested changes, and the manuscript has improved notably. However, there are still several adjustments that would further enhance the clarity and overall quality of the work. While these changes are not critical, addressing them is recommended before final acceptance.
- Introduction (Lines 61–70): The paragraph added contains relevant and valuable information. However, it is recommended to move the statement of the study’s aim to the end of the section, so that it naturally follows the background and rationale.
- Line 147: Please add the word “proteins” after caninum in the subheading.
- Results section:
- This section is currently divided into subsections whose subheadings mainly refer to the methodology (e.g., immunofluorescence, immunohistochemistry). It would improve readability if each subheading also summarized the main finding. The same approach is recommended for figure titles, making each figure as self-explanatory as possible.
- Example: The title of Figure 2 is “Histopathological and immunohistochemical findings in the brain of the infected mouse.” To make it self-explanatory, it could be rephrased as: “Histopathological and immunohistochemical findings in the brain of mice experimentally infected with Neospora caninum.”
- Figures 3 and 5: Please add “caninum” after “Neospora”.
- Figure 3: Please clarify whether the title should read “the infected mouse” or “infected mice”.
Author Response
Reviewer 3
Comments and Suggestions for Authors
The authors have incorporated the suggested changes, and the manuscript has improved notably. However, there are still several adjustments that would further enhance the clarity and overall quality of the work. While these changes are not critical, addressing them is recommended before final acceptance.
Response: Thank you very much for reviewing our revised manuscript (microorganisms-3777056). Your insightful comments have helped us improve our paper. To address the reviewer’s comments on the English, we requested an English editing service from MDPI Author Services and added this information to Acknowledgments section. Revisions made by the editing service are shown in blue, and further revisions made in response to the editor’s suggestions are shown in red. We are here submitting our revised manuscript entitled “The Distribution of Neospora caninum Secretory Proteins in Mouse and Calf Brains” for your kind consideration of its suitability for publication in Microorganisms. Our incorporation of the editor’s and reviewers’ suggestions are as follows.
Comment 1:
Introduction (Lines 61–70): The paragraph added contains relevant and valuable information. However, it is recommended to move the statement of the study’s aim to the end of the section, so that it naturally follows the background and rationale.
Response: Thank you for consideration and helpful comments on our revised manuscript. As your comment, we have revised the description of introduction.
Line 56-65.
However, the distribution of other secretory proteins, NcCYP, NcPF, and NcGRA6, in the brain of infected animals has not been researched. Notably, even for NcGRA7, no study has examined the relationship between the distributions of these secretory proteins and lesions in cattle. N. caninum surface antigen 1 (NcSAG1), non-secretory proteins abundantly expressed on the plasma membrane of tachyzoites, has been widely studied as a diagnostic marker [37-40], making it a useful reference for comparison with secretory proteins. In the present study, we focused on the distribution of NcCYP, NcPF, NcGRA6, and NcGRA7, along with NcSAG1, in the brains of experimentally infected mice and naturally infected calves, to highlight a significant gap in our understanding of the pathogenesis of N. caninum infection in its natural hosts.
Comment 2:
Line 147: Please add the word “proteins” after caninum in the subheading.
Response: Thank you for helpful comments on our revised manuscript. As your comment, we have added the word.
Line 96: 2.4. Production of polyclonal antibodies against N. caninum proteins.
Comment 3:
Results section: This section is currently divided into subsections whose subheadings mainly refer to the methodology (e.g., immunofluorescence, immunohistochemistry). It would improve readability if each subheading also summarized the main finding. The same approach is recommended for figure titles, making each figure as self-explanatory as possible. Example: The title of Figure 2 is “Histopathological and immunohistochemical findings in the brain of the infected mouse.” To make it self-explanatory, it could be rephrased as: “Histopathological and immunohistochemical findings in the brain of mice experimentally infected with Neospora caninum.”
Response: Thank you for consideration and helpful comments on our revised manuscript. As your comment, we have revised the subheadings in result section and figure titles.
Line 176: 3.1. Immunofluorescence revealed the distribution of N. caninum proteins in the infected HFF cells.
Line 190: 3.2. Histological and immunohistochemical analysis showed pathological lesions and the distribution of N. caninumproteins in the brains of experimentally infected mice.
Line 236: 3.3. Histological and immunohistochemical analysis showed pathological lesions and the distribution of N. caninumproteins in the brains of experimentally infected calves.
Line 184: Figure 1. The distribution of Neospora caninum proteins, NcSAG1, NcCYP, NcPF, NcGRA6 and NcGRA7, in HFF cells infected with N, caninum, analyzed by immunofluorescence.
Line 207: Figure 2. Histopathological and immunohistochemical findings in the brain of mice experimentally infected with Neospora caninum.
Line 227: Figure 3. (A-E) The distribution of Neospora caninum proteins in the parasites and host cells of mice experimentally infected with N. caninum. (F) A classification based on the immunohistochemical staining patterns: type A when only parasites were stained and type B when the parasites, parasitophorous vacuole space, and host cytoplasm were stained. (G) The proportions of type A (non-secretory pattern) and type B (secretory pattern) staining as observed in brain sections from five mice experimentally infected with N. caninum.
Line 249: Figure 4. Histopathological and immunohistochemical findings in the brain of calves naturally infected with Neospora caninum.
Line 263: Figure 5. (A-E) The distribution of Neospora caninum proteins in the parasites and host cells of calves naturally infected with N. caninum. (F) A classification based on the immunohistochemical staining patterns: type A when only parasites were stained and type B when the parasites, parasitophorous vacuole space, and host cytoplasm were stained. (G) The proportions of type A (non-secretory pattern) and type B (secretory pattern) were observed in brain sections from the two calves naturally infected with Neospora caninum.
Comment 4:
Result section: Figures 3 and 5: Please add “caninum” after “Neospora”.
Response:  Thank you for helpful comments on our revised manuscript. As your comment, we have revised the figure titles.
Line 227:
Figure 3. (A-E) The distribution of Neospora caninum proteins in the parasites and host cells of mice experimentally infected with N. caninum. (F) A classification based on the immunohistochemical staining patterns: type A when only parasites were stained and type B when the parasites, parasitophorous vacuole space, and host cytoplasm were stained. (G) The proportions of type A (non-secretory pattern) and type B (secretory pattern) staining as observed in brain sections from five mice experimentally infected with N. caninum.
Line 263:
Figure 5. (A-E) The distribution of Neospora caninum proteins in the parasites and host cells of calves naturally infected with N. caninum. (F) A classification based on the immunohistochemical staining patterns: type A when only parasites were stained and type B when the parasites, parasitophorous vacuole space, and host cytoplasm were stained. (G) The proportions of type A (non-secretory pattern) and type B (secretory pattern) were observed in brain sections from the two calves naturally infected with N. caninum.
Comment 5:
Result section: Figure 3: Please clarify whether the title should read “the infected mouse” or “infected mice”.
Response: Thank you for helpful comments on our revised manuscript. As your comment, we have revised the figure title.
Line 227:
Figure 3. (A-E) The distribution of Neospora caninum proteins in the parasites and host cells of mice experimentally infected with N. caninum. (F) A classification based on the immunohistochemical staining patterns: type A when only parasites were stained and type B when the parasites, parasitophorous vacuole space, and host cytoplasm were stained. (G) The proportions of type A (non-secretory pattern) and type B (secretory pattern) staining as observed in brain sections from five mice experimentally infected with N. caninum.
